# LAYER-WISE PRE-WEIGHT DECAY

## ABSTRACT

In deep learning, weight decay is a regularization mechanism been widely adopted to improve the generalization performance. Previously, a common understanding of the role of weight decay was that it contributes by pushing the model weights to approach 0 at each time step. However, our findings challenge this notion and argue the objective of weight decay is to make the weights approach the negative value of the update term instead of 0, thereby indicating a delay defect in certain steps that results in opposing penalties. In addition, we study the negative side effect of weight decay, revealing it will damage the inter-layer connectivity of the network while reducing weight magnitude. To address these issues, we first propose real-time weight decay (RWD) to fix the delay defect by penalizing both the weights and the gradients at each time step. Then, we advance the decay step before the update function as pre-weight decay (PWD) to mitigate the performance drop raised by the side effect. To further improve the general performance and enhance model robustness towards the decay rate, we finally introduce a layer-wise pre-weight decay to adjust the decay rate based on the layer index. Extensive analytical and comparative experiments demonstrate that the proposed *layer-wise pre-weight decay* (LPWD) (i) exhibits remarkable robustness to the decay rate, and (ii) significantly improves the generalization performance across various conditions.

## 1 INTRODUCTION

Weight decay (WD) has gained growing popularity in recent years as an effective regularization mechanism to improve the generalization performance of deep neural networks (Hanson & Pratt, 1988; Ergen et al., 2023; Stock et al., 2019). Weight decay is also frequently referred to L2 regularization since they are equivalent in the standard SGD optimizer (Loshchilov & Hutter, 2017).

Existing research endeavors have revealed the effect of weight decay (Loshchilov & Hutter, 2017; 2018; Alshammari et al., 2022; Neyshabur et al., 2014). Graves (2011) interpreted weight decay from the Bayesian perspective, thinking weight decay restricts model capacity by indicating a Gaussian prior over the model weights. Zhang et al. (2018) elucidated three different roles of weight decay. Xie et al. (2020) proposed insights into weight decay from the perspective of learning dynamics. In summary, there is a consensus that weight decay improves the generalization performance of a model by encouraging the weights to approach zero at each time step (Krogh & Hertz, 1991). However, our findings raise doubt about this interpretation and additionally reveal another flaw of traditional weight decay.

**WD has a delay defect.** WD penalizes the weights calculated from previous step, introducing a delay defect where WD actually pushes current weights to the negative value of the current gradient instead of 0. The delay defect will lead larger weights after penalizing when the four current factors (learning rate, gradient, weight decay rate and weights) meet certain conditions. This situation opposes the intended target of weight decay and ultimately leads to a deterioration in generalization performance.

**WD potentially impairs performance.** Wd reduces the output of each layer by scaling down the weights of those layers. However, due to the prevalent usage of numerous non-linear activation functions in deep neural networks, WD also potentially distorts the feature distribution of hidden layers. In other words, weight decay weakens the interconnections between network layers, ultimately resulting in a decline in performance.

Table 1: Classification results on both Cifar-10 and Cifar-100 datasets using ConvNext$_{tiny}$/SwinTransformer$_{tiny}$ and Adam/SGDM. We compare LPWD with Baseline (use neither), L2 ($L_2$ regularization), and WD. The mean and standard deviation of the best 10 test Top-1 accuracy during training are reported. The best settings of learning rate and weight decay rate for each method are searched in Table 2.

| | Cifar-10 | | | | Cifar-100 | | | |
| | ConvNext$_{tiny}$ | | SwinTransformer$_{tiny}$ | | ConvNext$_{tiny}$ | | SwinTransformer$_{tiny}$ | |
| Method | Adam | SGDM | Adam | SGDM | Adam | SGDM | Adam | SGDM |
|---|---|---|---|---|---|---|---|---|
| Baseline | 90.53±0.08 | 89.97±0.01 | 89.28±0.08 | 88.72±0.04 | 73.06±0.07 | 72.51±0.02 | 71.91±0.10 | 71.13±0.11 |
| L2 | 89.85±0.12 | 90.06±0.04 | 88.84±0.10 | 88.56±0.16 | 72.74±0.36 | 72.46±0.17 | 71.61±0.08 | 71.26±0.16 |
| WD | 90.77±0.05 | 90.13±0.02 | 89.47±0.10 | 88.74±0.09 | 73.88±0.22 | 72.81±0.03 | 72.07±0.10 | 71.42±0.10 |
| **LPWD (ours)** | **91.03±0.11** | **90.28±0.05** | **89.73±0.10** | **88.98±0.08** | **74.48±0.04** | **73.38±0.08** | **72.70±0.09** | **71.95±0.10** |

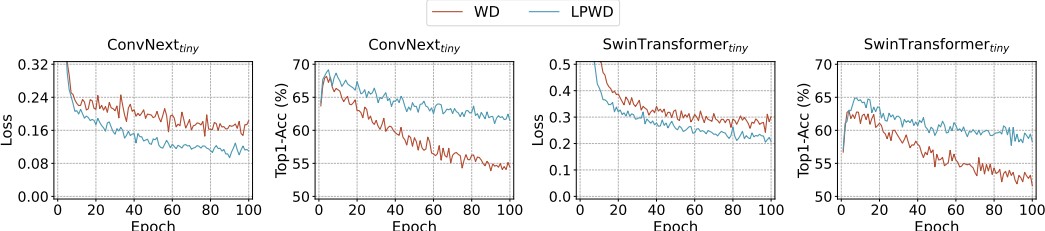

Figure 1: Training loss and test top1-accuracy comparison of LPWD and WD with strong penalty on cifar-100. Experiments are based on ConvNext$_{tiny}$/SwinTransformer$_{tiny}$ and Adam is adopted. The learning rate $\eta$ and weight decay rate $\lambda$ are set to 0.001 and 0.5 respectively.

In this paper, we propose a novel layer-wise pre-weight decay (LPWD) focusing on overcoming the delay defect, mitigating the performance drop raised by conventional weight decay, and further enhancing decay strategy through customizing decay strength for each layer. Specifically, to avoid opposing penalty, we first present real-time weight decay (RWD), which penalizes both the weights and the gradient. RWD promises that applying weight decay will always drive the weights smaller in magnitude. Then, we propose to decay before the learning phase as pre-weight decay (PWD). After the model is around converged, the learning phase will then help to mitigate the feature distortion through strengthening the connectivity between layers, reducing the performance drop additionally. For deep neural networks, high-level features possess fewer samples and more abstract semantic information than low-level features, leading to a greater risk of overfitting. Therefore, we finally introduce layer-wise pre-weight decay (LPWD) to customize the weight decay rate for each layer based on their index, imposing a weaker penalty for shallow layers and a stronger penalty for deep layers.

We conducted comprehensive analysis and experiments on various datasets (Cifar-10 and Cifar-100 (Krizhevsky et al., 2009)), using different optimizers (SGD (Loshchilov & Hutter, 2017) with momentum(SGDM) and Adam (Kingma & Ba, 2014)) and different the state of the art architectures (ConvNext (Liu et al., 2022) and Swin Transformer (Liu et al., 2021)). Our proposed LPWD consistently outperforms weight decay and other methods in all conditions, as listed in Table 1. In particular, when using a large learning rate and weight decay rate, LPWD exhibits even more significant improvements compared to tradition WD, as shown in Figure 1. This highlights the strong robustness of LPWD to weight decay rates.

## 2 FINDINGS

### 2.1 DELAY DEFECT

Previously, weight decay was wildly interpreted as a strategy to constrain the complexity of the model by encouraging the weights of the model to approach 0 gradually. However, our findings have uncovered a different reality. Weight decay can sometimes impose incorrect penalties due to a delay effect. For any given optimizer, the weight update function for weights $\theta$ at time step $t$ can be

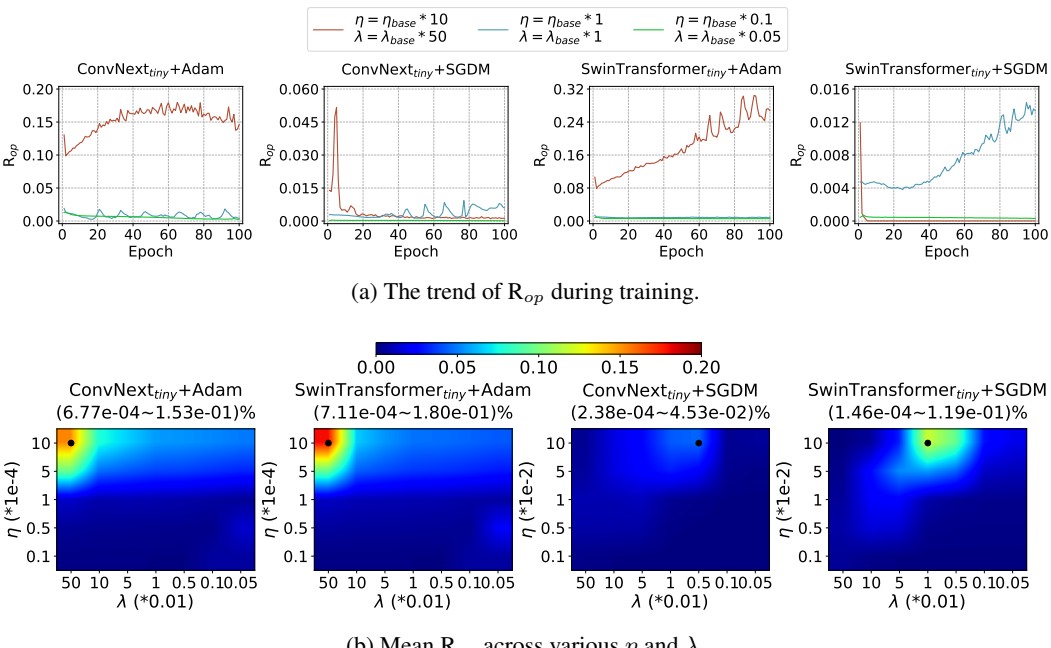

(a) The trend of $R_{op}$ during training.

(b) Mean $R_{op}$ across various $\eta$ and $\lambda$.

Figure 2: Investigation of the opposing penalty rate $R_{op}$ of WD. We conduct experiments on cfiar-100 using ConvNext$_{tiny}$/SwinTransformer$_{tiny}$ and Adam/SGDM. $\eta_{base}$ is set to $10^{-4}$ for Adam, $10^{-2}$ for SGDM, and $\lambda_{base}$ is set to $10^{-2}$ for both optimizers. The identification rule of opposing penalty is illustrated in section 4.1 and the detailed searching ranges of $\eta$ and $\lambda$ are introduced in Table 2. *(The black circle denotes the maximum of $R_{op}$)*

simplified as:

$$\theta_{t+1} = \theta_t - \eta_t U_t^g - \lambda\theta_t, \tag{1}$$

where $\lambda$ represents the weight decay rate, and $U_t^g$ is the update term calculated from the gradient, then multiplied by the learning rate $\eta_t$. In this case, weight decay performs the penalty based on the weights obtained from the previous step. Surprisingly, when $\lim_{\lambda\to1}\theta_{t+1} = -\eta_t U_t^g$, which contradicts initial expectations of the weight decay and may harm the generalization performance of the network. Specifically, consider the following inequality:

$$(\theta_t - \eta_t U_t^g - \lambda\theta_t)^2 > (\theta_t - \eta_t U_t^g)^2, \tag{2}$$

which can be simplified as:

$$\begin{cases} \eta_t U_t^g < (1 - \frac{\lambda}{2})\theta_t, & \text{if } \theta_t < 0, \\ \eta_t U_t^g > (1 - \frac{\lambda}{2})\theta_t, & \text{if } \theta_t > 0. \end{cases}$$

When $\eta_t$, $U_t^g$, $\lambda$ and $\theta_t$ satisfy the above conditions, WD will drive the current weights away from 0, resulting large weights compare to not apply it and potentially increasing the risk of overfitting. This is contrary to the initial exception of employing WD. We observed the mean rate of incorrect penalty during training, as shown in Figure 2. While the incorrect penalty rate is generally low, considering the large number of parameters in deep neural networks, this defect will impair generalization performance seriously. Similarly, the delay defect also exist in other regularization methods such as $L_2$ regularization, $L_1$ regularization, etc.

## 2.2 WEIGHT DECAY POTENTIALLY IMPAIRS PERFORMANCE

When incorporating weight decay with an optimizer, the update function at the time step $t$ can be separated into two independent sub-steps:

$$sub - step1 : \hat{\theta}_{t+1} = \theta_t - \eta_t U_t^g, \tag{3}$$
$$sub - step2 : \theta_{t+1} = \hat{\theta}_{t+1} - \lambda\theta_t, \tag{4}$$

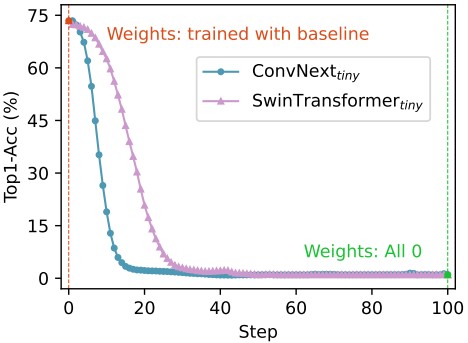
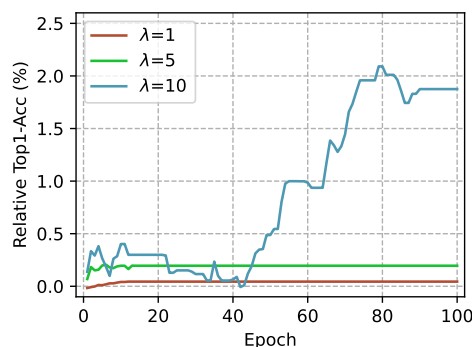

(a) Zero-shot weight decay using both ConvNext$_{tiny}$ and SwinTransformer$_{tiny}$ models trained on Cifar-100 adopting Baseline method. We decay model weights to 0 in 100 steps without extra training.

(b) Relative improvements of PWD over RWD (Acc$_{PWD}$-Acc$_{RWD}$) across vairous $\lambda$. Experiments are based on ConvNext$_{tiny}$ and Adam with a learning rate of 0.001. The mean of the best 10 test Top-1 accuracy during training is reported.

Figure 3: Subfigure (a) and (b) report the the results of zero-shot weight decay and RWD vs. PWD experiments respectively. (a) is related to Section 2.2 while (b) is related to Section 3.2.

where $\hat{\theta}_{t+1}$ represents the estimated next weights. Sub-step 1 aims to encourage the model to learn the current task, while Sub-step 2 is intended to scale down the weights in magnitude to limit the model's capacity and improve generalization. However, sub-step 2 has a side effect in that it can potentially disrupt the distribution when employing non-linear activation functions. Specifically, given a layer index $i$, the function of the $i_{th}$ layer to process the input $X_i$ can be defined as:

$$\hat{X}_{i+1} = \psi(W_i^\top X_i), \tag{5}$$

where $\psi(*)$ denotes the activation function such as GELU (Hendrycks & Gimpel, 2016) and sigmoid (Finney, 1947), and $W_i$ indicates the matrix consisting of weights and biases. After applying weight decay, this becomes:

$$X_{i+1} = \psi((1 - \lambda)W_i^\top X_i). \tag{6}$$

The application of $\psi(*)$ leads that $\hat{X}_{i+1}$ and $X_{i+1}$ are not linearly related: $\hat{X}_{i+1} \neq \frac{1}{1-\lambda}X_{i+1}$. This not only reduces the weight scale but also alters the internal relationships within the distribution of the input to the next layer. Since the cascading effects in deep neural networks, this change may further significantly harm performance. To visualize the adverse effects of weight decay comprehensively, we conduct zero-shot weight decay experiments. As shown in Figure 3a, accuracy consistently declines as the weight decay rate increases.

## 3 METHODS

In this section, we illustrate the proposed layer-wise pre-weight decay (LPWD) exhaustively. Specifically, we first design real-time weight decay (RWD) to fix the delay defect, then enhance it as pre-weight decay (PWD) to mitigate the performance drop raised by the side effect. We additionally propose to incorporate PWD with a layer-wise decay rate strategy (LPWD), which further boosts the general performance and improve model robustness towards the weight decay rate significantly.

### 3.1 REAL-TIME WEIGHT DECAY (RWD)

The delay defect usually arises from the potential incompatibility between the current weights and the forthcoming update term. To address this issue, we propose a RWD that simultaneously decays both the current weights and the update term. This can be mathematically expressed as:

$$\theta_{t+1} = \theta_t - \eta_t U_t^g - \lambda(\theta_t - \eta_t U_t^g). \tag{7}$$

This equation can also be regarded as the decay of $\hat{\theta}_{t+1}$ (from Equation 3), where $\theta_{t+1} = \hat{\theta}_{t+1} - \lambda\hat{\theta}_{t+1}$. RWD ensures decay in real time, such that, $\lim_{\lambda \to 1} \theta_t = 0$. Furthermore, it guarantees

$$(\theta_t - \eta_t U_t^g - \lambda(\theta_t - \eta_t U_t^g))^2 \leq (\theta_t - \eta_t U_t^g)^2 \tag{8}$$

always holds for $\lambda \in (0, 1]$. As a result, this straightforward approach promises to consistently drives the weights to be smaller at every time step if $\theta_t - \eta_t U_t^g \neq 0$, avoiding incorrect penalties during training completely. Algorithm 1 illustrate RWD in detail.

**Algorithm 1:**
**Real-time weight Decay (RWD)**

> **repeat**
> $\quad t \leftarrow t + 1$
> **Update Function:**
> $\quad g_t \leftarrow \nabla f_t(\theta_t)$
> $\quad U_t^g \leftarrow g_t$
> $\quad \hat{\theta}_{t+1} = \theta_t - \eta_t U_t^g$
> $\quad \theta_{t+1} = \hat{\theta}_{t+1} - \lambda \theta_t$
> **until** $\theta_{t+1}$ converged
> **return** $\theta_{t+1}$

**Algorithm 2:**
**Pre-weight Decay (PWD)**

> **repeat**
> $\quad t \leftarrow t + 1$
> $\quad \theta_t = \theta_t - \lambda \theta_t$
> **Update Function:**
> $\quad g_t \leftarrow \nabla f_t(\theta_t)$
> $\quad U_t^g \leftarrow g_t$
> $\quad \theta_{t+1} = \theta_t - \eta_t U_t^g$
> **until** $\theta_{t+1}$ converged
> **return** $\theta_{t+1}$

**Algorithm 3:**
**Layer-wise Pre-weight Decay (LPWD)**

> **Initial:**
> $\quad \lambda_i = \lambda \frac{i}{n}$
> **repeat**
> $\quad t \leftarrow t + 1$
> $\quad \theta_t = \theta_t - \lambda_i \theta_t$
> **Update Function:**
> $\quad g_t \leftarrow \nabla f_t(\theta_t)$
> $\quad U_t^g \leftarrow g_t$
> $\quad \theta_{t+1} = \theta_t - \eta_t U_t^g$
> **until** $\theta_{t+1}$ converged
> **return** $\theta_{t+1}$

## 3.2 PRE-WEIGHT DECAY (PWD)

While RWD effectively overcomes the delay defect, it still suffer from the side effects discussed in Section 2.2, especially when applying a large decay rate. To mitigate the performance drop associated with weight decay, we propose a PWD. As the PWD algorithm 2 illustrates, weight decay is applied before the update function. This means that after the model has converged, weight decay still affects the connectivity between layers, but the learning function (Equation 3) subsequently helps counteract this negative influence at each time step. However, experiments comparing RWD and PWD reveal that PWD significantly contributes to performance improvement only in cases where a large decay rate is utilized (as shown in Figure 3b).

## 3.3 LAYER-WISE PRE-WEIGHT DECAY (LPWD)

Given dataset $\mathcal{D} = \{x_s, y_s\}_{s=1}^N$, from the perspective of feature pyramid (Lin et al., 2017; Zhang et al., 2020), it can be reformulated as $\mathcal{D} = \{\{l_s^j\}_{j=1}^{a_s}, y_s\}_{s=1}^N$ or $\mathcal{D} = \{\{h_s^k\}_{k=1}^{b_s}, y_s\}_{s=1}^N$, where $l_s^j$ and $h_s^k$ represent the $j$-th low-level feature and the $k$-th high-level feature of sample $x_s$, respectively. In the feature extraction pipeline, low-level features such as color, shape and texture tend to have more samples than high-level features w.r.t semantic information such as a cat and a dog. This can be expressed as $a_s \gg b_s$. . Therefore, shallow layers are less risky to overfit compared to deep layers. To maximize the benefits of weight decay, we additionally propose to customize the weight decay rate for each layer based on layer index $i$ ($i \in [1, n]$). This process can be mathematically expressed as:

$$\lambda_i = \lambda \frac{i}{n}, \tag{9}$$

where $n$ is the total number of layers of the model. In this case, the penalty becomes stronger as $i$ increases. Moreover, since shallow layers are less sensitive to the initial decay rate, LPWD also shows strong robustness to the decay rate. As shown in Figure 1, LPWD demonstrates a much slighter performance drop than WD while the weight decay rate increases.

## 4 EMPIRICAL VALIDATION

We theoretically introduced the findings and our proposed LPWD in previous sections. In this section, we empirically demonstrate the findings and validate the advantages of LPWD over other methods comprehensively. we explore delay defects under various conditions and present delay defects that occur when using different models and optimizers in Section 4.1. Then we analyze the side effects of WD in Section 4.2, demonstrating it will impair the performance seriously especially. In Section 4.3, we compare LPWD with Baseline, $L_2$ regularization, and WD on different datasets, and the proposed method achieves remarkable improvements over other methods, and the gap is more noticeable

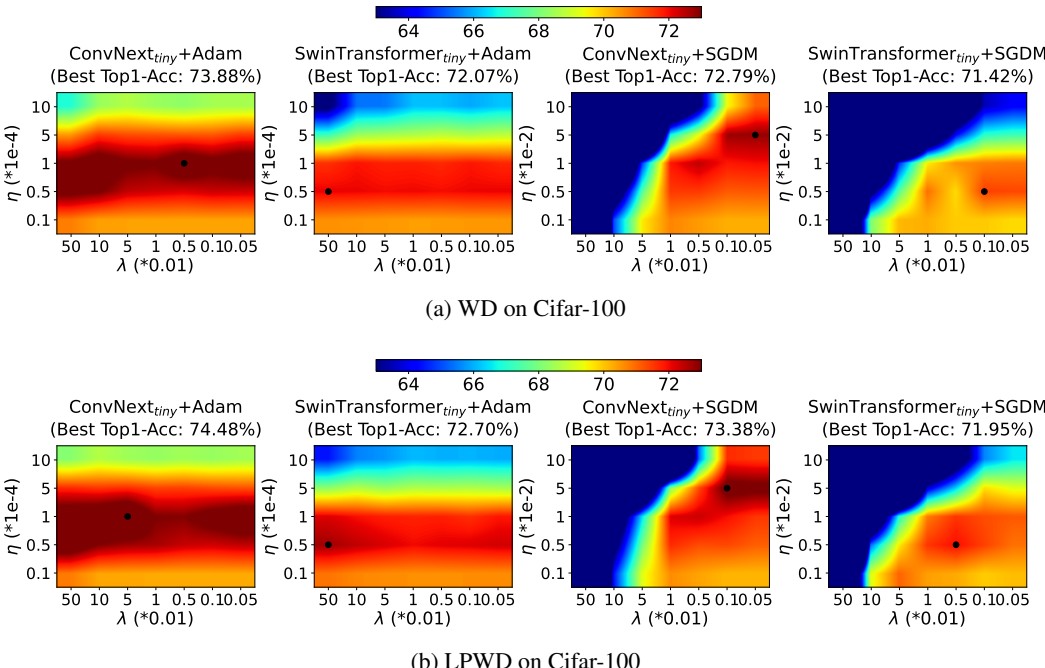

Figure 4: Detailed test results of LPWD and WD on Cifar-100 across various settings of learning rate $\eta$ and weight decay rate $\lambda$. ConvNext$_{tiny}$/SwinTransformer$_{tiny}$ and Adam/SGDM are adopted. The mean of the best 10 test Top-1 accuracy during training is adopted. *(The black circle denotes the best results.)*

especially when given a strong penalty. We additionally investigate LPWD at different model scales in Section 4.5, where it enjoys a more significant boost compared to WD as the model scale increases. Ultimately, we conduct ablation experiments in Section 4.6 presenting the contributions of each sub-method of LPWD in detail.

**Datasets and Augmentations.** All experiments are based on two popular datasets: Cifar-10 and Cifar-100 (Krizhevsky et al., 2009). For augmentations, HorizontalFlip with a probability of 0.5 from albumentations (Buslaev et al., 2020) is employed.

**Models and Optimizers.** We conduct experiments using the state-of-the-art models: ConvNext (CNN) (Liu et al., 2022) and SwinTransformer (Transformer) (Liu et al., 2021), both are load from timm library (Wightman, 2019) with ImageNet-21k (Russakovsky et al., 2015) pretrained weights. ConNext-tiny and Swintransformer-tiny are employed if there are no specific instructions. We chose SGD with momentum (SGDM) (Polyak, 1964) and Adam as the base optimizer, where the momentum factor of SGDM is set to 0.9 and Adam with ($\beta_1 = 0.9, \beta_2 = 0.999, \epsilon = 10^{-8}$) are adopted for all experiments.

**Implementation Details.** Images are normalized using ImageNet (Deng et al., 2009) default means and standard deviations. Image size and batch size are set to $32 \times 32$ and 512 respectively. We set a base learning rate $\eta_{base}$ of $10^{-2}$ for SGDM and $10^{-4}$ for Adam, and the base weight decay rate $\lambda_{base}$ is set to $10^{-2}$ for both optimizers. All $\eta_{base}$ and $\lambda_{base}$ are grid searched over ranges listed in Table 2. For all experiments, models are trained for 100 epochs with a constant $\eta$. The maximum of $\lambda_i$ for LPWD is set to $2\lambda$ to ensure $\frac{1}{n}\sum_{i=1}^{n}\lambda_i = \lambda$. To mitigate randomness, the mean of the best 10 Top-1 accuracy on the test set during training is adopted.

### 4.1 OPPOSING PENALTIES RAISED BY DELAY DEFECT

We first study the opposing penalty rate $R_{op}$ of WD been theoretically analyzed in Section 2.1. For weight $\theta_t^j$ at time step $t$, if

$$(\theta_t^j - \eta_t U_t^j - P^j)^2 > (\theta_t^j - \eta_t U_t^j)^2, \tag{10}$$

Table 2: Settings of learning rate $\eta$ and weight decay rate $\lambda$ for Adam and SGDM in most experiments. Various optimizers prefer different $\eta$.

| Hyper-parameters | Adam | SGDM |
|---|---|---|
| $\eta_{base}$ | 0.0001 | 0.01 |
| $\lambda_{base}$ | 0.01 | 0.01 |
| $\eta$ | {0.001, 0.0005, 0.0001, 0.00005, 0.00001} | {0.1, 0.05, 0.01, 0.005, 0.001} |
| $\lambda$ | {0.5, 0.1, 0.05, 0.01, 0.005, 0.001, 0.0005} | {0.5, 0.1, 0.05, 0.01, 0.005, 0.001, 0.0005} |

Table 3: Ablation studies on Cifar-100 using ConvNext$_{tiny}$ and Adam. We ablate each method with the corresponding optimal settings searched over Table 2. The mean and standard deviation of the best 10 test Top-1 accuracy during training are reported.

| Method | WD | RWD | PWD | LPWD | Top-1 Acc (%) |
|---|---|---|---|---|---|
| Baseline | ✓ | | | | 73.88±0.22 |
| Baseline | | ✓ | | | 74.19±0.08 |
| Baseline | | | ✓ | | 74.29±0.11 |
| Baseline | | | | ✓ | **74.48±0.04** |

the penalty term $P^j$ applied to $\theta_t^j$ will be marked as opposing penalty $P_o^j$, $R_{op}$ is defined as:

$$R_{op} = \frac{\sum P_o^j}{\sum P^j + \sum P_o^j}. \tag{11}$$

As shown in Figure 2b, the delay defect exists in different types of models and optimizers, and the mean $R_{op}$ for each combination of model and optimizer constantly increases as $\eta$ and $\lambda$ growing, but it's low overall during training. However, according to the cascading effect, it may still matter significantly in deep neural networks. Figure 2a shows the changes of $R_{op}$ during training using various $\eta$ and $\lambda$, where $R_{op}$ doesn't present a definite changing rule during training.

### 4.2 ZERO-SHOT WEIGHT DECAY

We investigate the impact of the side effect mentioned in Section 2.2 by applying penalties of varying intensity. When a large learning rate $\eta$ or decay rate $\lambda$ is given, the loss will lead to None. To avoid such a dilemma, We propose to only conduct decay steps to study the relationship between the side effect and the weight decay rate $\lambda_s$ comprehensively, where $s$ indicates step index. Specifically, the weights $\theta_s$ of the model are scheduled to decay to 0 in 100 steps,

$$\theta_s = \theta_0 - \frac{\lambda_s}{100}\theta_0. \tag{12}$$

As shown in Figure 3a, for both ConvNext and SwinTransformer models, the accuracy constantly decreases as $\lambda_s$ grows, presenting the side effects that exist in different types of models and will seriously impair the general performance especially when giving a large $\lambda_s$.

### 4.3 COMPARISON OF LPWD WITH OTHER METHODS

In this section, we empirically evaluate the performance of our method across various conditions. We compare LPWD with baseline, $L_2$ regularization, and WD using different networks and optimizers. As listed in Table 1, LPWD achieves state-of-the-art generalization performances with a significant improvement compared to other methods. Figure 4 presents the detailed test results of LPWD and WD, where the proposed method outperforms WD in many cases, demonstrating a more robust hyper-parameter space.

### 4.4 EXAMINING LPWD UNDER STRONG PENALTY

In practical applications, $\eta$ and $\lambda$ are often not set to be optimal (hyper-parameter exporting can be extremely computationally expensive), indicating a robust regularization mechanism is crucial. In addition, we examine the sensitivity of LPWD towards the weight decay rate. We directly turn to a large penalty since a small one doesn't reflect noticeable differences. Given large $\eta$ and $\lambda$ (weight

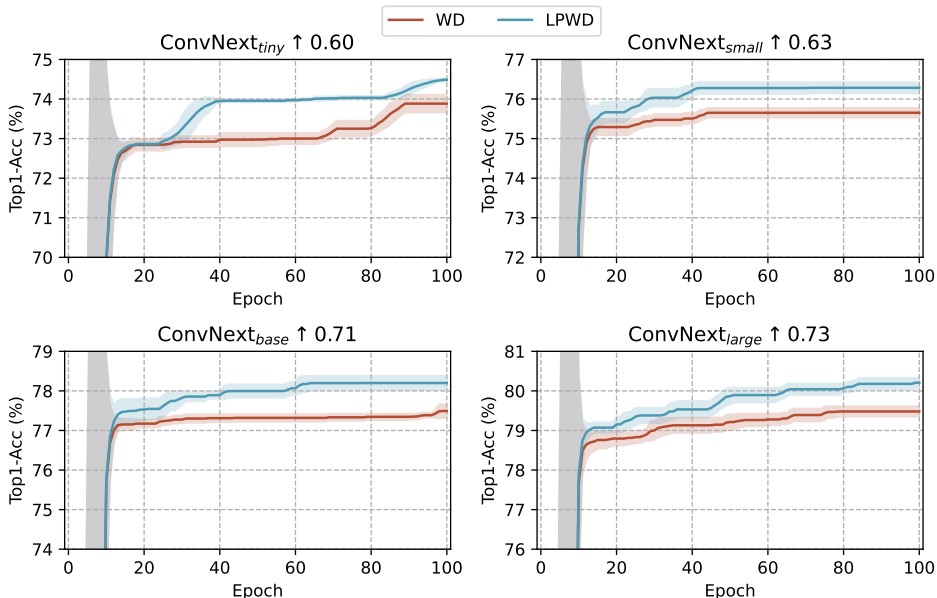

Figure 5: comparison of WD vs. LPWD at different model scales using ConvNext series models and Adam optimizer. The curve denotes the mean of the best 10 test Top-1 accuracy during training and the shadow indicates the related standard deviation.

decay term is multiply by $\eta$ and $\lambda$), as shown in Figure 1, the loss of LPWD constantly lower than WD while the test top1-accuracy of LPWD constantly higher than WD, both with significant gap, demonstrating LPWD exhibits stronger robustness towards weight decay rate than WD through different conditions.

## 4.5 EXPLORING LPWD AT DIFFERENT MODEL SCALES

Recently, large models have gained growing popularity. Since previous experiments were based on tiny models with less computation, we experimented with LPWD on larger models. We compare LPWD with WD at different model scales to explore the influence in detail. As shown in Figure 5, LPWD demonstrates more significant improvements compared to WD as the model scale increases. It can be attributed to the cascading effect inherent in deep neural networks, where the negative consequences of delay defects and side effects tend to amplify as the depth of the network increases.

## 4.6 ABLATION STUDY

In Section 3, we propose and analyze the advantages of the RWD, PWD, and LPWD. To verify the importance of the proposed method, this section shows detailed comparison and ablation studies. As listed in Table 3, RWD contributes most significantly, a Top-1 accuracy improvement of around 0.31%, while PWD helps minimally (the $\eta$ and $\lambda$ are small), and LPWD generally boosts the performance further. However, the contribution of PWD can be exceptionally impressive when large $\eta$ and $\lambda$ are given, as shown in Figure 3b, the mean of the best test Top-1 accuracy of larger $\lambda$ is significantly higher than smaller $\lambda$.

## 5 CONCLUSION

Our theoretical analysis reveals that traditional weight decay (i) holds a delay defect, which leads to opposing penalties in certain steps, and (ii) distorts the distribution of features where the related layer cooperates with a non-linear activation function.

To avoid delay defects, we suggest RWD that decays both the weights and the gradients at each time step. Then, we introduce a PWD based on RWD to mitigate the feature distortion by advancing the decay step ahead of the update function. Finally, we propose LPWD to combine layer-wise weight

decay rate strategy to customize decay rate based on layer index from the perspective of feature pyramid.

Extensive experiments on various datasets using different models and optimizers demonstrate that LPWD (i) shows strong robustness towards decay rate, (ii) significantly improves the generalization performance compared to other methods across various conditions.

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
