# OpenReview forum: "Layer-wise Pre-weight Decay"
_ICLR.cc/2024/Conference — ICLR 2024 Conference Withdrawn Submission_

### Official Review · Reviewer_QiZQ · 2023-10-16

**Soundness:** 2 fair
**Presentation:** 2 fair
**Contribution:** 2 fair
**Rating:** 3
**Confidence:** 4

**Summary:**

The authors study the role of weight decay (wd) in deep learning. They argue that WD can be harmful since 1) the WD update and gradient update are calculated in parallel, and 2) the nonlinearities in the networks mean that the output from nonlinear layers is not scaled down linearly. The authors propose a few variants of WD. 1) RWD (eq. 7) where the WD terms account for the gradient updated, 2) PWD where the WD update seems to be computed before the gradient update, and 3) PWD where the WD coefficient changes per layer. Experimentally, the authors consider finetuning SWIN and ConvNext models from TIMM on Cifar datasets. The authors generally find that their proposed methods lead to small improvements. Some ablation experiments are also provided.

**Strengths:**

- WD is used everywhere, so improving it would have a significant impact.
- The idea is natural and easy to understand.

**Weaknesses:**

- The paper is not very well written. E.g. the authors write “was wildly interpreted” – they probably mean widely. There are also technical things that are not clear. E.g. section 2 is very handwavy and I don’t understand what the authors mean. I would encourage them to formulate their thoughts as mathematical theorems with proofs. The algorithms are also unclear.
- The metrics reported by the paper seem strange. The authors state that “To mitigate randomness, the mean of the best 10 Top-1 accuracy on the test set during training is adopted”. Firstly, it seems like this is in effect doing model selection via the test-set – a separate validation set should be used for that. Secondly, the different scores during a training run and not independent, so it’s a little misleading to use these to calculate error bars. Thirdly, it seems like each experiment is only done once. So we might just be looking at statistical noise.



# Minor issues:

The authors state that “weight decay weakens the interconnections between network layers, ultimately resulting in a decline in performance”. This is not true in general. If it was, people would not use WD.

The authors state that “For deep neural networks, high-level features possess fewer samples” – this is true for CNNs where the feature maps shrinks, but not for e.g. ViTs.

**Questions:**

1. Could you do more runs and use error bars from independent runs?

2. Has layerwise coefficients for WD been proposed in previous papers? If so, which?

3. What is the difference between RWD and PWD? To me, it looks like they will just alternate gradient and WD updates.

4. It seems like you only provide finetuning experiments. Can you provide pretraining experiments?

---

### Official Review · Reviewer_AMDG · 2023-10-28

**Soundness:** 2 fair
**Presentation:** 2 fair
**Contribution:** 2 fair
**Rating:** 3
**Confidence:** 4

**Summary:**

The paper introduces a novel method called LPWD to address the shortcomings of traditional weight decay in deep learning models. The authors identify two main issues with traditional weight decay: (1) a delay defect that leads to opposing penalties in certain steps, and (2) distortion of feature distribution when used with non-linear activation functions. To address these issues, the authors propose RWD (Real-time Weight Decay) and PWD (Progressive Weight Decay). LPWD combines these methods with a layer-wise weight decay rate strategy. Empirical validation on datasets like Cifar-10 and Cifar-100 using models such as ConvNext and SwinTransformer demonstrates the robustness and improved generalization of LPWD over traditional methods.

**Strengths:**

In general, I like the idea of LPWD and look forward to see its potential in more evaluations.

1. LPWD consistently outperforms other methods, especially in scenarios with strong penalties or larger model scales.
2. The proposed method shows strong robustness towards decay rate, making it more reliable in diverse scenarios.

**Weaknesses:**

My concerns are around the limited evaluation. It might lead to wrong motivation and conclusion that could be fatal errors.

1. The empirical validation is primarily based on Cifar-10 and Cifar-100. Testing on more diverse datasets might provide a clearer picture of LPWD's effectiveness. Also, the improvements are subtle.
2. If the authors only tested LPWD during fine-tuning, it might limit the generalizability of their claims. The behavior and benefits of LPWD during the entire training process (from scratch) might be different.
3. Fine-tuning on a specific task with a pre-trained model might introduce some bias, as the model has already learned general features from the pre-training dataset. This could affect how weight decay or any regularization method operates.
---
minor:
Typo: The delay defect will lead larger weights after penalizing when the four current factors (learning rate, gradient, weight decay rate and weights) meet certain conditions. “Lead to”?

**Questions:**

1. How does LPWD perform on larger datasets like ImageNet or more complex tasks like object detection or segmentation?
2. How does LPWD perform for training from scratch?

---

### Official Review · Reviewer_51Q7 · 2023-10-29

**Soundness:** 3 good
**Presentation:** 2 fair
**Contribution:** 3 good
**Rating:** 6
**Confidence:** 3

**Summary:**

This paper identifies the two issues of common weight decay including the delay effect and damaging the inter-layer connectivity. Then it proposes the RWD and PWD to address it. It finally proposes a combined version with a layerwise index scaling trick as an approach and demonstrates improved results compared to baseline WD in ConvNext and ViT.

**Strengths:**

* The problem studied is novel, fundamental, and important for deep neural network optimization.
* The proposed method is neat and the effect of each solution is experimentally studied. The significance of each solution in a combined approach is carefully compared.
* The experiment setup and evaluation are solid, and the improvement is pronounced for an optimization method.

**Weaknesses:**

* The paper indeed uses clear mathematical notation to illustrate the problem, but I don't think the two issues are theoretically identified. In particular, there is no claim or proof to show how the issue directly affects generalization. I think the way the author identifies the issue is by pointing out the extreme case in which the issue can fail, e.g. (2), or a very large decay rate. Also, through empirical study of Figure 2 and Figure 3.
*  The second issue and the solution lack clarity, It is not clear how Figure 3a illustrates the issue of layer-wise connectivity and how the proposed method solves it.
*  The paper is a bit hard to follow because the mathematical illustration of the issues and empirical demonstration are separated (I don't think the mathematical representation is enough to identify the issue), which makes Figure 2 on Page 3 first introduced and described on Page 7, and Figure 1 on Page 2 introduced on Page 8.
*  Should the Algorithm 1 the red line be $\theta_{t+1}=\hat{\theta}_{t+1}-\lambda \hat{\theta_t}$?

**Questions:**

* If the searched hyperparameters learning rate and weight decay coefficient are 5 x 7 as shown in Table 2, how could the 2D figures in Figure 4 be so smooth?

**Details Of Ethics Concerns:**

No ethics concern.

---

### Official Review · Reviewer_hMNw · 2023-10-30

**Soundness:** 1 poor
**Presentation:** 2 fair
**Contribution:** 1 poor
**Rating:** 3
**Confidence:** 5

**Summary:**

The paper claims to have two key observations about the drawbacks of weight decay, i.e., delay defect and feature distortion. To address the two issues, the paper proposes real-time weight decay (RWD) and layer-wise pre-weight decay (LPWD). The effectiveness of the proposed weight decay is shown on CIFAR10 and CIFAR100 with both CNN and Transformer.

**Strengths:**

The main observations and proposed methods are highlighted, which makes the paper well-organized.

**Weaknesses:**

The reason why the two major findings hurt the performance is not clear. The first finding is that WD will sometimes drive the current weight away from 0. The paper claims that as a result of the large number of parameters and cascading effect of deep neural networks, this phenomenon will hurt generalization. The second finding is that WD distorts the distribution of features, and again due to the cascading effect, this phenomenon will hurt the performance.

This cascading effect of deep neural networks is used to explain all the effects in this paper, without specific reasoning process and verifiable arguments. Thus, the findings do not quite make sense to me as it is hard to see how the drawbacks of WD affect the performance.

**Questions:**

I don't quite understand the performance gap between RWD and PWD. If one considers the iterative optimization process, the two methods are virtually the same except for at the first step and final step. The performance gain of PWD in Fig. 3b is probably due to learning rate decay at epoch 40.

What is the difference between RWD/PWD and decoupled weight decay [1]?

[1] Loshchilov, Ilya, and Frank Hutter. "Decoupled Weight Decay Regularization." International Conference on Learning Representations. 2018.

---

### Meta-Review · Area_Chair_ZoR7 · 2023-12-02

**Metareview:**

This paper studies some negative effects of weight decay.
- A delay effect that pushes weights to the negative direction to the gradient direction.
- Distortion of feature distributions when used with non-linear activation functions, which weakens the inter-layer connectivity.

Then, they proposed real-time weight decay (RWD) and layer-wise pre-weight decay (LPWD) to address these issues. Empirical results on CIFAR10, CIFAR100 with ConvNext, and Vision Transformers are shown to evaluate the proposed methods.

**Justification For Why Not Higher Score:**

- Lack of clear explanations of the observed findings on weight decay. There is no understanding of the direct effect of WW on generalization.
- Limited empirical validations and datasets. Limited improvements using the proposed method.
- Limited empirical evaluations only during fine-tuning but not pretraining.
- Some technical discussions are handwavy. Some grammatical issues and typos.
- Non-standard way of measuring the performance on the test sets.

**Justification For Why Not Lower Score:**

N/A

---

### Decision · Program_Chairs · 2024-01-16

Reject